# Mining Corporations, Democratic Meddling, and Environmental Justice in South Africa

**Llewellyn Leonard** 

Department of Environmental Science, College of Agriculture and Environmental Sciences,
University of South Africa, Science Campus, Florida 1709, South Africa; llewel@unisa.ac.za; Tel.: +27-11-471-2311

**Abstract:** During Apartheid, the mining industry operated without restraint and compromised the ecology, the health of mining workers, and local communities. The lines between the mining industry and government was often unclear with the former influencing government decisions to favour uncontrolled operations. Although new post-Apartheid regulations were designed to control negative mining impacts, the mining industry and the state still have a close relationship. Limited academic research has empirically examined how mining corporations influence democracy in South Africa. Through empirical investigation focusing on Dullstroom, Mpumalanga and St. Lucia, KwaZulu-Natal, this paper examines how mining corporations, directly and indirectly, influence democratic processes at the macro state and micro community levels. At the macro level, this includes examining mining companies influencing government decision-making and enforcement to hold mines accountable for non-compliance. At the micro level, the paper examines mining companies influencing democratic processes at the local community level to get mining developments approved. Findings reveal that political connections between the mining industry and government, including collusion between mining corporations and local community leadership, have influenced mining approval and development, whilst excluding local communities from decision-making processes. Industrial manipulation has also influenced government in holding corporations accountable. This has contributed towards not fully addressing citizen concerns over mining development. Democracy in post-Apartheid South Africa, especially for mining development is, therefore, understood in the narrow sense and exposures the realities of the ruling party embracing capitalism. Despite challenges, civil society may provide the avenue for upholding democratic values to counter mining domination and for an enabling political settlement environment.

**Keywords:** mining; democracy; environmental justice; government; South Africa

## 1. Introduction

During Apartheid, the mining industry was virtually immune to effective environmental regulation and operated freely and without restriction (Hallowes and Munnik 2006; Leonard 2017a). The separation of powers between mining corporate conglomerates and the government was often indistinct (Van Wyk et al. 2009). Apartheid gave industries licences to undertake enormous environmental destruction, which involved harming the ecology, the well-being of employees and surrounding communities (Fig 2005). The end of Apartheid brought some confidence that fresh legislation would also force industry to act within the required regulation and constrain the pollution, which had generated severe environmental and health problems (Leonard 2009). For example, the 1996 Constitution of the Republic of South Africa makes provision for a right to a healthy environment, and the right to have the environment protected by preventing pollution and degradation (South African Constitution 1996). The National Waste Management Strategy (1998) aims to reduce the generation and environmental impact of all forms of waste and to ensure the health of the people

and the quality of the environment. The Mineral and Petroleum Resources Development Act (2002) was formulated for new government involvement in the minerals economy to deal with the history of biased policies that barred the mainstream populace from involvement in mining development (Rogerson 2011). This paper focuses on the mining industry since although mining has been the main source of the country's wealth; it has resulted in substantial collapse to black communities and has propagated disunity amongst white and non-white groups (Mzwakali 2017). According to Mabuza et al. (2010), mining companies have strategically taken advantage of weak local government or political upheaval within communities to influence governance and local decision-making over development. However, limited academic research has empirically examined how mining corporations actually influence democracy in South Africa.

According to Khan (2003: in (Vaughn and Ryan 2006)) South African companies generally have also been extremely reluctant to reveal which political parties they have funded, and the primary sources of political funding remain unknown. This acquired industrial funding generates a potential conflict of interest despite strong legislation. As Hallowes and Munnik (2006) note, local government officials in South Africa are wined, banqueted and reminded of business input to government proceeds. This questions the influence that large corporations (especially mining) have on democracy and social actors (i.e., government and civil society) to act in their favour. For example, although South African legislation has tried to improve corporate liability for corporate activities post-1994, government bodies have not managed corporate governance (Malherbe and Segal 2001). For example, from September 2012 to November 2013 46 mines were operating without water use licenses (Davis 2013). It is uncertain why the national Department of Water Affairs (DWA) did not hold these mining companies liable for non-compliance. According to the South African environmental justice human rights organisation groundWork (2003), government has consistently failed to implement compliance in terms of existing rules and permits through prosecution or active sanctions, choosing to negotiate with industry the terms of sustained non-compliance. It is therefore questionable how negotiation may unfold between government and industry and on whose terms. As Castells (2000) notes for political interaction in the 'network society' where policy and developmental practices are progressively being designed not only by the state, but also by other interest groups in the network (e.g., corporate and civil society interests).

Since democracy, communities have been progressively outspoken in their disapproval to mining development and the level of pollution impacts on their health and environment. Several civil society groups in a letter to the Acting Chairperson of the Portfolio Committee on Mineral Resources dated 6 September 2013 noted mining companies and the Department of Mineral Resources (DMR) lack of participation and notification of interested and affected parties (IAPs) about mining development decisions. This also included a lack of participation and notification to the proprietors and inhabitants of land on which mining was planned—that a right had been approved (even when requested to do so). Other concerns highlighted rejections by the DMR and the consultants appointed by mining companies to provide IAPs with information about applications (e.g., proposed provision for rehabilitation, financial and technical information). It is doubtful if the state is also limiting progressive environmental regulation to offer a conducive environment to enhance its vision for industrial policy. As Fig (2005) highlights, surrounding the state's efforts to limit environmental regulation, suggested modifications in the law, covering required Environmental Impact Assessments (EIAs) is generally viewed by government as a technique to streamline development, but by opponents as an effort to weaken environmental standards to favour industry. As Leonard (2017b) notes for mining development and EIA's in South Africa, mining applications have been approved by government without mining companies having conducted proper public participation processes or sharing relevant information about the development process, questioning how government makes informed decisions for mining development without a proper EIA process having unfolded. Although South African mining is considered important for the economy, it contributes the second least to job creation and after the

utilities sector (Refer to Table 1). In 2017, the mining sector contributed eight percent to the countries nominal Gross Domestic Product (GDP)[1] (Refer to Figure 1).

**Table 1.** A comparison of South African mining direct jobs creation compared to other sectors.

| Industry | April–June 2017 | January–March 2018 | April–June 2018 |
|---|---|---|---|
| Total (Thousand) | 16,100 | 16,378 | 16,288 |
| Agriculture | 835 | 847 | 843 |
| Mining | 434 | 397 | 435 |
| Manufacturing | 1799 | 1849 | 1744 |
| Utilities | 148 | 143 | 161 |
| Construction | 1395 | 1431 | 1476 |
| Trade | 3265 | 3276 | 3219 |
| Transport | 954 | 960 | 1014 |
| Finance and other business services | 2395 | 2402 | 2399 |
| Community and social services | 3560 | 3785 | 3692 |
| Private households | 1311 | 1275 | 1296 |

Source (Statistics South Africa 2018).

**Figure 1.** Contribution of the mining sector to the South African GDP compared to other sectors. Source (Brand South Africa 2018).

Within the above context, this paper is interested in the extent that mining corporations have influenced government (and civil society) interests for mining development and hence democratic processes. This paper focuses on the mining industry since not only is it the central source of the country's wealth (next to tourism) but also due to the contestation over mining and the serious

---

1　The GDP is a monetary measure of the market value for the final total of goods and services produced in a period. Nominal GDP includes changes in prices due to inflation or a rise in the overall price level.

social and environmental problems it causes. The Centre for Environmental Rights (CER), a public interest law group, reported on thirty companies (mostly mining) listed on the Johannesburg Stock Exchange (JSE) to identify companies that integrated the principles of the triple bottom line—environmental, social and economic sustainability. The CER found in their 'Full Disclosure' reports (Centre for Environmental Rights 2015, 2016), that most companies were causing severe environmental abuses and were not disclosing this sufficiently to shareholders. This also included a lack of standard reporting requirements for environmental compliance. This paper, therefore, includes examining mining companies influencing government decision-making and government enforcement of regulation to hold mines accountable for non-compliance. The paper will also examine mining companies influencing democratic processes at the local community level to get mining developments approved. Through empirical investigation focusing on Dullstroom, Mpumalanga and St. Lucia, KwaZulu-Natal, this paper examines the power of mining corporations to influence democratic processes at the macro and micro levels. The paper firstly examines the literature surrounding policy, governance and the power of mining corporations before exploring corruption and corporate influence on democracy and state decision-making. Mining companies influencing local community interactions will then be examined before discussion the literature surrounding democratic and political settlement epistemologies. The methodology employed will then be presented followed by the results, discussion, conclusion and recommendations.

*1.1. Macro-Economic Policy, Governance and the Power of Mining Corporations*

The democratic South African transition influenced the continuation of corporate power to favour mining development. During the first years of change to democracy, corporations were worried of deep-seated financial change, causing business delegations to visit ANC leaders in exile from the late 1980's, to create mutual understanding (i.e., corporate continuity of economic and ecological authority and deregulation) (Fig 2005). According to Madihlaba (2002), the start of negotiations resulted in the mining industry and government looking at the need for reforms. A meeting in Zambia led by the Chief Executive Officer of Anglo American resulted in a series of meetings between business and the exiled ANC (Levy et al. 2015). Spicer (2016) notes how corporations engaged intensively with all the political parties in numerous consultations, conferences and seminars on a prospective economic policy. This included a number of scenario planning exercises carried out building on the Anglo-American scenarios of the mid/late 1980s, which had helped shape thinking of some National Party elites during Apartheid. Thus, upon assuming power in 1994, the ANC, in exile a socialist movement, embraced capitalism (Malherbe and Segal 2001). Pellow (2006) refers to market economies (capitalism) as continually creating ecological and social harm through increasing rates of production and consumption, with Gould et al. (2004) noting capitalism as increasing corporate profitability at the expense of workers, their social benefits (i.e., wages) and the environment. Monshipouri et al. (2003) notes that some aspects that add to the upsurge in the authority of corporations include global shield of their property and their noticeable part in global bodies that control trade, such as the World Trade Organization (WTO), which results in the growing power of corporations in the context of international trade. This results in governments being constrained in their capacity to offer social safety nets and public services to cushion the negative consequences of engaging in a global environment.

In contemporary South Africa, the lack of environmental governance and integration of environmental concerns into development processes has allowed mining corporations to conduct business as usual and retain power. According to Fig (2005), despite the regulatory roles of the state being established, features of environmental governance remain weak. Regulation of the environment is disjointed with functions divided among the national departments, the nine provincial departments, and the specialized regulatory bodies. Government has politically given limited attention to the environment, and there is failure to assimilate environmental anxieties into mainstream planning, development and macroeconomic policy. The lack of government technical expertise has also resulted in the government at times relying on industrial expertise to maintain its regulative function. For example,

according to the South African Chemical Workers Industrial Union, scientific experts utilise their skills to provide a service to the chemical industry as contractors. Some of these experts are former employees of the very industry they are contracted to observe (Parliamentary Report 2005). Government cannot ensure a fair decision over industrial development and operations if government relies on scientists associated with those institutions to inform decisions. As Hallowes and Munnik (2006) note, regulation is highly uneven between the different local areas due to capacity constraints and disputed between corporations wanting self-regulation and ordinary citizens demanding effective official regulation. According to Fig (2005), industry, particularly in the minerals, energy and chemical sectors, have been unwilling to obey with new legislation, knowing that government enforcement is feeble. Leonard (2017a, 2017b) also notes that the South African Department of Mineral Resources (DMR) has been reluctant to work with other government departments (i.e., Environment) to hold mining companies accountable for non-compliance since it has been pro-mining and more concerned with social and economic development. Hallowes and Butler (2002) also suggested that, since Apartheid, industry in South Africa has tried to convince government that industry should sidestep new constricting regulations and support voluntary measures and self-regulation as a means to maintaining profit.

### 1.2. Corruption and Corporate Influence on Democracy and State Decision-Making

Corporate influence over the state has swayed state decision-making. For South Africa specifically, according to the Organization for Economic Co-operation and Development (2014) regarding the Anti-bribery Convention in South Africa, corruption is still a major concern in the country with mining identified as one of the high-risk sectors. The country has generally had numerous prominent domestic corruption scandal and accusations (i.e., contracts being granted due to enticement, special contacts or bureaucrats also holding business interests) that are connected to the uppermost rankings within government. Unscrupulous government leaders are appointed in 5–10 year mind-sets and this creates issues around how much fortune they can accrue while in office (Mzwakali 2017). For example, South Africa's elite police fraud investigation (i.e., the Hawks) raided the DMR offices in 2011 surrounding a corruption investigation over the alleged issuance of rights to Imperial Crown Trading's (Mine Web 2011). Pillay (2007) notes that in certain instances coal mining companies with political black-economic empowerment partners have also used these contacts to their advantage when applying for mining applications. Thus, civil society and local community groups have accused the DMR for favouring mining applications with political connections.

Mining influence over government and state decision-making has also been evident in other countries. According to Aulby (2017) in Australia, mining conglomerates expend huge sums of money on lobbying and political donations to sway policy decisions in their favour. The mining industry is 80 percent foreign-owned and has spent over $541 million over a decade lobbying Australian governments. The income to the mining lobby is funded by corporate affiliation payments and these memberships are dominated by foreign owned businesses (e.g., Peabody, Anglo American, BHP, Rio Tinto, Glencore and Adani Mining). In the present discussion over coal mines and the involvement of environmental groups, the mining lobby has requested two key policy changes, which include cancelling the tax deductibility standing of environmental NGOs and prohibition of foreign political contributions to third parties, including environment and social groups. Smith (2014) also notes that when corporations donate funds to political parties, there is an understanding (rarely made explicit) that large campaign donations buy political access and favourable consideration in policy development and legislation. According to Human Rights Watch (2011), there have been failures by Toronto-based Barrick Gold mining company to address the social and environmental effects of its operations. Papua New Guinea has plentiful natural resources, but weak governance and corruption have prohibited this wealth from helping citizens. Canada's government has not implemented important oversight or regulation of the overseas operations of Canadian companies. Bill C-300 was formulated to levy

improved government oversight but was beaten in Canada's House of Commons in October 2010 since Barrick petitioned forcefully against the measure.

### 1.3. Mining Companies Influencing Local Community Cohesion against Mining Impacts

Mining conglomerates may take advantage of poor communities to push mining development in their favour whilst also dividing locals. According to Mzwakali (2017), mining-affected communities are generally poor, with high rates of unemployment and they are divided over mining development. Politicians, traditional leaders and businesses for their own benefit normally capture land ownership and mining community trusts. Madihlaba (2002) noted that in 1994 in Clewer, a small community near Witbank, Mpumalanga experienced serious coal mining issues, with a mining operation opening less than thirty meters away from residential homes. Mine personnel attempted to create friction between the community and the mineworkers by claiming to the trade union that residents had intimidated mineworkers and prevented them from going to work. This was proven untrue. Mine officials also sowed discord between the community and miners over job losses should the mine close. However, mining companies have also attempted to influence local community cohesion in other countries around the world. For example, Lopez and McDonagh (2017) note that Canadian corporation Pacific Rim Mining arrived in El Salvador in 2002 to extract gold located in the Cabanas region of the Lempa River basin. Although the local community and social movement successfully organised against the mining operations, the mining company initially engaged in strategies to infiltrate the local community to support mining development. The company set up the El Dorado Foundation in the Cabanas region. It then implemented various local projects to present themselves as a benefactor of 'development' to the communities. The foundation acted as the social arm of the company in order to recruit people sympathetic to mining. Pacific Rim also promised jobs to people to win their support, generating false expectations in the population. Gustafsson (2017) noted for mining in Peru, whilst corporations may offer some groups with basic services, these relationships have also resulted in fragmented interests and forms of demand that are disconnected from democratic rights, questioning how private politics affect broader processes of democratic deepening in Peru and elsewhere in Latin America (including for South Africa).

### 1.4. Democracy and Political Settlement Epistemologies

Before proceeding, it is important to understand what national democracy entails and how corporations may influence democracy. The idea is not to engage in detailed discussions but to get a sense of what the overall democratic framework necessitates. The term democracy has multiple interpretations. Hay (2005), democracy involves political parties mobilising public opinion, possibly guaranteeing that policy processes shadow the considered preference of the public sentiment. Scholte (2001) refers to democracy as a community of people exercising collective self-determination. According to Post (2006), democratic forms of government are the same people to whom they apply making the laws. From the definitions presented, democracy appears to involve participatory supremacy, consultation, transparency and open agreement. Cohen and Arato (1992) refer to the elite model of democracy (i.e., voting a political party into power) and the participatory model of democracy (i.e., including citizens in decision-making processes). They claim that with the elite model of democracy, there is no guarantee that citizens set the political agenda, make political decisions, or choose policies. As Greenstein (2003) notes, since democracy in South Africa, little attention has been given to transforming how the state power is organised, dispersed and implemented within, and how it intermingles with civil society. This suggests that South Africa falls into the elite model of democracy. The participatory model is similar to the savage theory of Claude Lefort referred to by Nasstrom (2006) in that the logic of representation originates in the change of power from the representatives to the represented. Savage democracy activates a process of circularity between society and political institutions and excludes political delegation. According to Keane (1998) within democratic states,

civil society and the state function as separate but interdependent internal articulations of a process in which power is subject to public disputation, compromise and agreement.

There is a strong link between citizen activism and democracy. According to Naidoo (2003), there is a misconception that citizen activism undermines democratic processes by 'short-circuiting' recognized procedures for decision-making. Active citizenry is essential for a healthy democratic society, complements democratic practices, and provides a constant check on official accountability. The local level also contributes to innovative ideas and knowledge resulting in ownership within communities. Involvement and discussion are important fundamentals in the process of collective decision-making and hence democracy. According to Vitale (2006), 'deliberative democracy' acknowledge citizens as being the foremost actors in the political process with a strong ideal of involvement. Political decision-making occurs in a framework of broad public discussion, in which all participants can discuss the numerous concerns in a cautious and sensible manner for democratic legitimacy to occur. Thus, the deliberative aspect resembles a shared procedure of reflection and examination, pervaded by the dialogue that heads the result. Discourse and democracy are thus according to Vitale two sides of the same coin. Once legally institutionalized, the dialogue principle transforms into the principle of democracy. Both share a common source, since all political power has to extract from the unrestrained authority of the citizens. As Cavanagh and Mander (2004) note, the principle of subsidiarity respects the notion that power resides in people. Legitimate authority flows upwards from the populace through the expression of their democratic will. According to Post (2006), people need to engage in decisions that affect them and values of self-government are essential to democracy. Self-government is about the authorship of decisions, not about the making of decisions. However, engagement in neoliberal and market forces may influence national democracy imperatives. As Monshipouri et al. (2003) note, free trade and its procedures, or lack thereof, are inadequate to encourage a level playing field for democracy and therefore there is a need for greater social responsibility of corporations considering their increasing influence over privatisation and their acquired powers, which have been traditionally vested only in governments. Unfortunately, confronted with burdens to secure international investments, governments in the South have limited or no alternative but to be open to the positions of international corporations, thus compromising democratic values.

It is useful to link discussions of democracy to political settlement theory as a way to reach an agreement. There have been many definitions about what political settlement entails. These include how power is distributed or balanced between contending social groups and classes (Di John and Putzel 2009; Behuria et al. 2017) to understanding the formal and informal processes, agreements and practices that help consolidate politics rather than violence for dealing with disagreements and with the distribution of power (Kelsall 2016). For Menocal (2015) political settlements outline the parameters of inclusion and exclusion in a political system in terms of process (i.e., who is included in decision-making) and outcome (e.g., how wealth is distributed). However, common throughout the definitions is that political settlement is about the distribution of power and about inclusion, with Behuria et al. (2017) highlighting that the distribution of power evolves constantly in a political settlement. However, Desai and Woolcock (2012) stress the importance of removing obstacles to effective and sustained participation (e.g., communication, education, awareness) and for political settlement to occur, with Di John and Putzel (2009) emphasising the need for the distribution of rights and entitlements across groups and classes on which the settlement is based. Thus, political settlements offer a robust focus on the role that capitalism and political organisations play in shaping development processes and outcomes (Hickey 2013) and hence it is useful for uncovering unequal settlements and as a hindrance for democratic consolidation.

## 2. Method

The study adopted a qualitative approach, which entailed semi-structured interviews with key informants. For the data analysis, grounded theory and open coding were used to recognize similar themes across the interviews. This aided in breaking the data into appropriate groups. This article

focuses on the theme of (1). Political connections and corruption between mining companies and government and (2). Mining companies exploiting vulnerable communities to spearhead mining development (discussed below). Secondary information was also used to verify and/or substantiate interviewee responses where possible and find relations or patterns with the interview content. Fieldwork to explore mining corporations influencing democracy was part of a larger study conducted in Dullstroom, Mpumalanga in 2013 and St. Lucia, KwaZulu-Natal in 2016 and 2017. These specific study sites lack academic research on mining development. Whilst research has been conducted for larger scale mining sites and for mining sites such as in Marikana due to political upheaval and the killing of mining workers by state police (see for example Marinovich 2016; Hill and Maroun 2015) this paper focuses on the lesser well known and smaller mining sites that also deserve attention.

## 2.1. Participants

In Dullstroom, personal interviews were conducted with Dullstroom landowners, community leaders/representatives, farmers, local youth organisations, public legal institutions, external environmental NGOs, local/provincial government departments and the mining industry. Sixteen interviews were conducted of which six interviews are used for this article (i.e., Interviewees A–F below). A resident contacted in the Dullstroom area was secured from an academic colleague who had previously researched tourism employment in the area. A snowballing technique was employed to secure other informants and also secured from contacts found in secondary documents such as press releases. Regrettably attempts to secure interviews from key personnel within the DMR (i.e., Regional Manager Mpumalanga), including the Department of Water Affairs (DWA) (i.e., Chief Director: Legal Services) proofed unsuccessful due to no responses received. In 2016, personal interviews were conducted in Fuleni with two local community leaders fighting mining development, one woman anti-mining activists and one youth leader. Informants in this area were secured through media article contacts and by using a snowballing technique during data collection. Unfortunately, several attempts in 2016 to get interviews from key personnel such as the Secretariat Regional Mining Development and Environmental Committee (RMDEC) KwaZulu-Natal, Co-operative Governance and Traditional Affairs (COGTA), KwaZulu-Natal Wildlife, Naledi Development Consulting (the Fuleni mining consultants) and the Chief Executive Officer from Hluhluwe-iMfolozi game reserve—proved fruitless due to no responses received. In 2017, several interviews were conducted for the Somkele fieldwork as part of a larger study into mining and water security, which included a group interview comprising of twelve individuals (mostly youth). Unfortunately, two residential interviewees who work for the local Somkele mining company initially agreed to be interviewed but provided they received compensation for their time. Since the author could not agree to this due to ethical reasons, the interviews did not proceed. One traditional leader initially agreed to be interviewed but later changed his mind. For the St. Lucia overall, a further two telephonic interviews were conducted with non-residential social actors (i.e., external non-governmental organization) assisting the local Fuleni and Somkele communities fight against mining development (Interviewee K) and with a public legal institution supporting community anti-mining activists (Interviewee H). For St. Lucia (Fuleni and Somkele), six interviews are used for this article (i.e., interviewees F–K). In total eleven interviews are reported in this paper.

## 2.2. Procedure

Before the interviews, information was provided to participants surrounding the purpose of the study and informed consent was obtained from participants. All the interviews conducted were recorded on a digital recorder and later transcribed. Information was then manually sorted into codes and themes were generated. Considering that mining is contentious in South Africa, with threats and assassination of community members speaking out against mining (see Ntongana 2018), the author has anonymised all interviewees despite informants noting that their names be used for research purposes.

## 3. Results and Analysis

*3.1. Political Connections and Corruption between Mining Companies and Government*

Some interviewees suggested that there were links between mining companies and national government (i.e., DMR), with mining companies influencing government decision-making over mining development. This, in turn, limited governance over mining development and allowed mining applications to be approved without taking informed decisions. An anonymous Dullstroom biodiversity interviewee A (Interview, 4 October 2013) from a public interest organisation, indicated that many of the mining companies were politically connected with important government officials and this ensured that these mining companies could dictate on their own terms how they operated. According to another Dullstroom interviewee B (Interview, 4 October 2013) from the Escarpment Environment Protection Group interested in biodiversity protection:

> " . . . he's [mining applicant] is connected with somebody within the government or within the DMR, I can promise you . . . So if they [government] now start saying no, no, no, no Steenskampsburg is out I'm not going to allow you to do it [mine], they going to be pissing off somebody [mining connection] . . . We were at the cuff of having such a meeting in 2008 [to proclaim a protected area] . . . where agriculture, ourselves, Environmental Affairs, Water Affairs, DMR, labour, Premier's office, everybody would sit around the table and draw up these maps and identify priority areas and it was cancelled two days before. There is no political will to do it . . . they all said we agree with you 100% but if you implement it I will be without a job"

It was noted by some interviewees that mining companies, because of political connections, were being assisted by national government to get their mining applications approved, suggested control by industry over government decision making. According to interviewee C, an Environmental Scientist at a public interest organisation (Interview, 5 October 2013) and a resident of Dullstroom noting for the Regional Mining Development and Environmental Committee (RMDEC) committee, which oversees mining applications. The proposed Mineral and Petroleum Resources Development Amendment Act (2008) only makes provision for government representatives to be appointed to the RMDEC by the Minister (although inputs can be heard from civil society) and which committee is responsible for making informed decisions to advise the minister on mining developments. According to the MTPA interviewee C, government representatives on the RMDEC were also assisting mining companies with mining applications to get the applications approved:

> "She [mining applicant] applied for that farm [for coal mining] . . . but . . . there were I think something like 11 inconsistencies . . . I objected on all of it . . . The applicant . . . [had] high standing in the politics and her husband was a mining engineer. I didn't know that . . . he [DMR regional manager] was fighting me so that they [could] . . . override my objection . . . She [applicant] said . . . the people here, in this [DMR] office . . . assisted her with her application . . . They all work together . . . to get prospecting and mining rights . . . If you do an application you must have an independent consultant."

By mining companies and government having a close working relationship, this undermined how the parameters of inclusion and exclusion were outlined. This influenced political settlement. This resulted in the exclusion of local residents to be included in discussions over mining development in their area. Power was unfairly distributed to government and especially mining corporations to the exclusion of other stakeholders. Unfortunately, many mining companies did not use independent mining consultants during applications. Some mining consultants had also worked closely with government entities. According to a Dullstroom resident interviewee D (Interview, 5 October 2013) who had placed a mining application to mine coal on his farm:

> "The reason we used him [environmental consultant] is because . . . he wrote most of the rules of water for Water Affairs . . . they [government] consult with him . . . "

This suggests that the mining applicant strategically used the consultant who not only understood the legislative loopholes but also due to close connections within government and which possibly influenced government decisions. As a report on combating corruption in mining approvals conducted by Caripis (2017) noted for South Africa, most government authorities do not have the capacity to confirm the contents of environmental and social impacts assessments (ESIAs), with the DMR accountable for approving ESIAs and issuing environmental authorisations. However, the Department does not have the required capability and knowledge to do this and has failed to accomplish its environmental duties. The report also found that there may be not accuracy or truthfulness in decisions surrounding ESIAs and the principles for awarding mining licenses will not be publicly knowable. This suggests that the DMR cannot make decisions on its own but may be reliant on external industry expertise. More power thus rested with mining corporations influencing democratic processes. Nevertheless, Caripis (2017) also found that due to the DMR being under-resourced this led to corruption risk with lengthy delays and many companies suing the DMR under the Promotion of Administrative Justice Act 2000 in order to achieve compliance with timelines, and even facilitating bribes. Therefore, the DMR in an attempt to avoid legal action against not being able to meet timelines may be blindly approving applications to avoid litigation. Unfortunately, mining companies were also found in the field to have strategically employed government officials by offering them higher salaries since they understood the mining applications process and could, therefore, assist them in getting mining applications approved faster. Some interviewees noted that senior government officials leave the DMR due to mining companies offering them higher salary packages. As biodiversity specialist, interviewee A noted (Interview, 4 October 2013):

" ... *Mining companies are employing these guys [from national government] to help them further—they offer them twice the salary ... and once he is employed by mining he knows the process. You get your license ... so much easier.*"

In addition to political connections between mining companies and national government, there was also corruption and bribery taking place at the local government level. As the Director at Dullstroom Trout Farm, Interviewee D (Interview, 3 October 2013) highlighted:

" ... *the local politicians they in on the act too ... deals are done so almost certainly the ... miners ... have certain important political partners—that's what you do ... [Government's excuse for allowing mining is that] we create jobs ...* "

According to a 2011 benchmarking study on 'Municipalities and Communities in Mining Initiatives' conducted by the Eastern Cape Socio-Economic Consultative Council (2011), the Middleburg local municipality was instrumental in the opening of a mine by Metorex Mining Corporation. This mine was partly funded by the Industrial Development Corporation (IDC) and pays royalties to the local municipality for every tonne mined, thus questioning practices of bribery/corruption and if royalties paid was an incentive for the municipality opening the mine (Industrial Development Corporation 2016). The IDC is a national development finance institution that promotes economic growth and industrial development (in line with the ANC's agenda of macroeconomic growth). The South African government under the supervision of the Economic Development Department owns it. The website of the IDC shows its board members composed of some individuals from the mining industry (e.g., Bobby Godsell previously served as the Chief Executive Officer (CEO) and Director of Anglogold Ashanti Limited and Deputy Chair and CEO of AngloGold America). Mining companies via the IDC may thus be having undue influence over mining development, questioning how mining developments unfold at local levels. Unfortunately, as was witnessed during Apartheid where there was lack of separation of powers between mining corporate conglomerates and the government, it seems that mining industries still have some influence over how mining development occurs in post-Apartheid South African society by having influence over government decision-making processes. This has further affected how political settlement has

unfolded and for democracy. The link between mining operations and local level corruption is held by Knutsen et al. (2017) referring to the 'resource curse' of having natural resources that ultimately perpetuate local corruption. The authors geographically matched 92,762 respondents from four Afro barometer waves with data on 496 mines in 33 African countries by reviewing the precise location and opening of industrial mines and the ways they influenced corruption locally. Micro-level evidence was that mineral extraction affected local-level corruption, with a positive correlation between corruption and active mines. The core finding was that mining actually increases local corruption.

### 3.2. Mining Companies Exploiting Vulnerable Communities to Spearhead Mining Development

Mining corporations were strategically manipulating vulnerable communities by promising residents' jobs to get mining developments approved. Several interviewees noted that since families were poor and unemployment was high, they were more willing to support mining development since corporations promised job creation from mining. According to Environmental Scientist interviewee C (Interview, 5 October 2013), noting for the strategy used by mining companies to exploit vulnerable communities:

> "... The mining companies are very clever ... most of them are ... overseas companies that have been developed for over many years, so they have got a very clever strategy ... So when they come to a community like this [Dullstroom township] it is so easy for them to exploit the innocence ... people are not very knowledgeable about what the end results are and what it actually all entails. So they [mining corporations] can easily exploit places in Africa ... I have seen it in my job that they do a lot of window dressing, fantastic promises and lovely plans, but when it comes to implementation they get away with murder ..."

Due to poverty, vulnerable communities, therefore, were not concerned about protecting their natural heritage and the environment. Rather, they tried to secure mining jobs to combat poverty. As the Environmental Scientist (Interview, 5 October 2013) further noted surrounding why it was easy for corporations to have an influence on poor communities:

> "Currently with the economics in South Africa ... a lot of people are unemployed, but then they don't think about things like aesthetical values and sense of place, they think about survival, so for those guys if you offer them jobs [to] mine here in the middle of Dullstroom they will take it ... some of them have got families that are starving, so we can understand that but unfortunately the mining companies abuses that situation for their own [benefits]"

This finding was also confirmed in the Somkele community in St. Lucia, KwaZulu-Natal, by youth leader J (Interview, 7 July 2017) where due to poverty and unemployment in the community, the local mining operation took advantage of this situation by promising more local jobs and bursaries for tertiary education, with the latter not materialising, whilst the former was miniscule. Generally, this suggests that mining companies were indirectly influencing local democratic process for communities to independently make decisions. Mining companies, therefore, indirectly influenced the inclusion and exclusion of residents over mining development decisions by promising jobs and social benefits. Mining companies, therefore, took advantage of local circumstances to influence democracy and political settlement. Due to divergent class differences in communities, mining corporations also caused local community fracturing across class (and race) lines. This was noted for Dullstroom when poor township residents in support of mining for job creation confronted residents opposing mining. This suggests that political settlement and democracy may be influenced by internal community differences and fragmentation. According to a local Dullstroom farmer, interviewee E (Interview, 5 October 2013):

> "[Pro-miner X—a Dullstroom farmer conducting mining on his property] was talking to the [township] people [about] job creation [and] they must go and have a strike in town ... and they can make millions out of this ... they [township residents] wanted to close the whole town ..."

A mining corporation also secured transport for those supporting mining to get to (anti-mining) meetings. Anonymous interviewee A:

" . . . *We had a stakeholder workshop where we wanted to present the intent to proclaim this area [Wakkerstoom, a protected area] . . . There were like 2000 or 5000 people who were bussed by mining [from the township] . . . They were protesting against our intent to proclaim it as a nature reserve . . . They [township residents] kicked chairs, they fought . . . they spoke about a figure of like 5000 [mining jobs] . . . I think shipped them by bus [by the mining company] . . . it's a dirty game actually that is being played . . . there is a politically well-connected member of this mining team. He actually went to go see the communities and I don't know if he paid for it or how it happened but he later when we were sitting around a table admitted that he may have spoken to the community . . .* "

According to an interviewee B (Interview, 4 October 2013) fighting against mining, noting for how mining corporations had encouraged local township residents to act against him specifically:

" . . . *the mining issue was an induced issue that was from a mining company that wanted to open pits at the back here [in Dullstroom] . . . they go into a township and buy off some of the councillors in the area and get a hundred people together and they start organising marches and those things. But it is money flowing in for persons, I mean they're virtually bought . . . in Chrissiesmeer . . . there was a public meeting . . . in came 300 or 400 locals, placards waving . . . my name was on those signs and [they were saying] we don't want greed and [interviewee X] . . . so I got up and walked between the middle of them and said whose this [interviewee X] . . . they have no idea and they were talking to me, so they were obviously bought . . . But when you unemployed and you don't have much of a hope of being employed in the future. If someone comes around and says you know we will give you employment but you must get rid of . . . these guys who are giving us a hassle . . . So it is very easy for a mining company to go and buy a little bit of influence in the township*"

Similarly, for other proposed mining sites in South Africa, in Fuleni, KwaZulu-Natal, mining companies engage in collusion with most traditional leaders to get mining developments approved despite opposition from local residents and hence influenced political settlement. Traditional leadership structures embraced mining development since they secured benefits from the industry. The leaders, however, supported mining in secret from the local community. An Interviewee F (personal interview, 2016) leader of the Ubumbano Youth Organisation (UYO), which organisation aims to unite the youth of Fuleni for local sustainable economic development, noted largely where the traditional leadership structures loyalty lied:

" . . . *there are people [traditional leaders] who are supporting the mine though they are not doing it openly . . . they [mining company] used to go to those individuals especially stakeholders [the traditional council] . . . who are connected to the mine . . . They [traditional council] are on the mine's side . . . not on the side of the community . . .* "

Whilst the earlier assertion noted by Mzwakali (2017) that mining propagates disunity amongst white and non-white groups (as evident above for Dullstroom), the case for Fuleni also shows that mining can also create intra-class and racial disunity. Several informants noted the support for mining development within the council was because of corruption as the traditional leaders profited from mining activities, with the mining corporation strategically targeting traditional leaders to ensure mining support. According to interviewee H (telephonic interview, July 2016) from a public interest legal firm specialising in environmental law and environmental justice and representing the Fuleni community in opposing the application made to the DMR for an open cast coalmine—noting for corruption between traditional leaders over mining development:

" . . . *it happens in all of these places where the traditional leaders end up being bribed quite substantially by the mining companies. It's quite cleverly done such that suddenly they are all driving new cars . . . so it's not in the form of hard cash . . . they are benefiting to be on the side of the mine so the whole community is anti the mine . . .* "

The above finding was also found in the Somkele community in KwaZulu-Natal. According to a Somkele youth leader interviewee G (Interview, 7 July 2017), mining corruption has occurred where the local mine has given money to local community leaders, with the latter being 'hungry lions where they feed themselves and forget what the community needs'. Nevertheless, corruption in Fuleni at the local leadership level extended beyond the traditional leadership council. Three interviewees (i.e., interviewee I and J (personal interview, 22 July 2016)—local Fuelni community resident leaders fighting against mining development and interviewee K (telephonic interview, 28 July 2017) a non-profit group supporting anti-mining residents) noted that the local ward councillor leader supported mining development due to family educational rewards received. This resulted in the ward leader not supporting local community anti-mining members. As the Fuleni anti-mining community interviewee I (personal interview, 22 July 2016), noted for how the local ward councillor leader was supporting mining due to corruption:

> *"Our ward councillor is on the side of the mine. His son has already been taken [by the mine] and they trained him in a couple of courses. So already, if the mine starts he is going to have a position in that company. So it was easy for him [ward councillor] to push for the mine to start because he will get something on his side ..."*

## 4. Discussion and Conclusions

Whereas during Apartheid, the mining industry operated without restraint and had undue influence over government decision-making to create an enabling operational environment to maximise profits at the expense of people and the environment, this practise is still present in post-Apartheid South Africa, although to a lesser degree. Firstly, government's engagement in a macro-economic development framework during the democratic transition has prioritised industrial expansion at the expense of social and environmental risks and resulting in poor environmental governance. This has influenced how political settlement has unfolded with little attention given to transforming how the state (and mining) power is organised, dispersed, and implemented, and how it intersects with civil society for democratic legitimacy. Results revealed that the mining corporations and government (national and local) having a close relationship, with the former entity able to influence the latter on how mining development occurred. Corruption was, however, an issue within government which resulted in poor governance over mining applications (i.e., as witnessed in Dullstrom) with government being under-resourced and with mining corporations strategically employing government officials so as to get mining development approved. Thus, new post-Apartheid South African regulations, designed to control the negative impacts of mining, have not been as effective due the influence and connections that mining corporations have with the enforcer (i.e., government). Interestingly, due to a democratic transition and the insertion of democratic and human rights principles introduced in the 1996 South African Constitution and into South African policies, laws and regulations, this has resulted in communities having a level of freedom of expression against government and corporate malpractice not witnessed during Apartheid. As a result, mining corporations have extended beyond the state to also influence local leaders (i.e., evident in St. Lucia) to get developments approved and to bypass proper consultation with local communities. Mining corporations also abused the poor socio-economic status of poor communities by using mining jobs to create local divisions (i.e., as evident in Dullstrom), thereby indirectly influencing the terms of inclusion and exclusion for democratic legitimacy for decisions over mining development. Thus, whereas during Apartheid and during the transition, mining corporations influenced government decisions over mining development, in contemporary South Africa mining corporations have now acted to interfere in local community democratic structures over mining development.

Democracy in post-Apartheid South Africa, especially for mining development is therefore understood in the narrow sense and exposures the realities of the ruling party embracing capitalism. The current notion and representation of civil society in South Africa and the implementation of a participatory model of democracy and its power to enable participatory forms of democracy by

government (and local community leadership) is questionable due to corporate interference. Local communities are not consulted meaningfully during mining development process. There is thus a need to trigger savage democracy with a transition of power to ordinary people. Leonard (2012) notes that there is a need for civil society leaders to also engage in transparency and take direction from the grassroots. However, Neocosmos (2011) notes, with the insertion of neoliberal frameworks into government settings, what is required is for the state to be modified in enabling a practice of justice to guarantee better social inclusion for the majority of citizens. This will also require that local communities and ordinary citizens hold politicians and local leader's accountable and by removing leaders if necessary. For now, it seems that civil society solidarity and strategies employed against mining companies and governments may be the best prospect for dealing with irresponsible mining companies and corrupt relations between mining companies, government and untrustworthy local leaders. Despite some challenges with South African civil society formations (see Leonard 2014) civil society does act as a new force in society against domination (Matten 2004) as witnessed in many local community struggles over mining development and corruption (see for example Furlong 2017; Leonard and Lebogang 2018; World Wide Fund 2011). These groups gain political momentum and broad support from the visible gap between the demands of the citizenry and their representation in the spectrum of government and local leadership (Beck 1992) and therefore provides the best potential for an enabling political settlement and for deliberative democracy. However, it remains to be seen if the ruling party government leadership continues to be dictated to by corporations and how civil society and local communities are able to organize and respond to any risks over mining development, including how local leaders will engage with their constituencies over any mining development decisions made.

**Funding:** This research received no external funding.

**Acknowledgments:** The author would like to thank all informants who participated in this study.

**Conflicts of Interest:** The author declares no conflict of interest.

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
