# Peer review of "Mining Corporations, Democratic Meddling, and Environmental Justice in South Africa"

_socsci, doi:10.3390/socsci7120259_

Round 1
Reviewer 1 Report
I think the paper is now suitable for publication.
Reviewer 2 Report
The author(s) addressed the changes and the paper can be accepted.
This manuscript is a resubmission of an earlier submission. The following is a list of the peer review reports and author responses from that submission.
Round 1
Reviewer 1 Report
Attached.

Author Response
REVIEWER 1 - Comment | Author Response |
The methodology has been nicely written but can the author(s) provide a brief justification why those cases (communities) were chosen? | Done, please refer to the methodology where this has been clarified (i.e. These specific study sites were chosen due to a lack of academic research conducted on mining development in these areas. Whilst limited research has been conducted in well-known sites such as Marikana due to political upheaval and the killing of mining workers by state police (see for example Marinovich 2016; Hill and Maroun 2015) this paper focuses on the lesser well known microsites that also deserve attention.) |
The author(s) discusses democracy and how it has not assisted South Africa to protect the environment and locals from the mining companies due to a seeming alliance between the national elites, local leaders and the transnational companies. This paper will be enriched if the author(s) discusses, (at least have a paragraph or so) on political settlement and how this shape resource use and development, even in democratic setting. For instance, di John and Putzel (2009, p.4) defined political settlement as the balance or distribution of power between contending social groups and social classes…’. It involves establishing a dominant political order and trying to persuade other competing political interests to accept the dominant view either through co-option, forced/suppression or elimination. Political settlements – that is, processes of intra-elite bargaining and negotiation (Desai and Woolcock 2012). Driven from a political economy perspective, the political settlements approach offers a strong focus on the role that capitalism and political organisations play in shaping development processes and outcomes (Hickey 2013, p. 6 – 7).
This list of references might help. Hickey, Sam (2013). Thinking about the politics of inclusive development: towards a relational approach. ESID Working Paper No. 1, University of Manchester.
Desai, D. and Woolcock, M. (2012). The Politics of Rule of Law Systems in Developmental States: ‘Political Settlements’ as a Basis for Promoting Effective Justice Institutions for Marginalized Groups. ESID Working Paper No. 8, University of Manchester.
Di John, J. and Putzel, J. (2009). ‘Political Settlements: Issues Paper’. Governance and Social Development Resource Centre, University of Birmingham, June 2009. | Literature on political settlement and linked to democracy has been added. Refer to last paragraph under section: Democracy and political settlement epistemologies
The following references have also been added:
Behuria, Pritish, Buur Lars and Gray Hazel. 2017. Studying political settlements in Africa. African Affairs 116(464):508-525
Desai, Deval and Woolcock, Michael. 2012. The Politics of Rule of Law Systems in Developmental States: ‘Political Settlements’ as a Basis for Promoting Effective Justice Institutions for Marginalized Groups. ESID Working Paper No. 8, University of Manchester. Available online: http://www.effective-states.org/wp-content/uploads/working_papers/final-pdfs/esid_wp_08_desai-woolcock.pdf. (accessed 21 November 2018)
Di John, Jonathan and Putzel, James. 2009. ‘Political Settlements: Issues Paper’. Governance and Social Development Resource Centre, University of Birmingham, June 2009. Available online: http://epapers.bham.ac.uk/645/1/EIRS7.pdf. (accessed 21 November 2018).
Hickey, Sam. 2013. Thinking about the politics of inclusive development: towards a relational approach. ESID Working Paper No. 1, University of Manchester. Available online: http://www.effective-states.org/wp-content/uploads/working_papers/final-pdfs/esid_wp_01_hickey.pdf. (accessed 21 November 2018).
Kelsall, Tim. 2016. Thinking and working with political settlements. Overseas Development Institute Briefing, January 2016. Available online: https://www.odi.org/sites/odi.org.uk/files/odi-assets/publications-opinion-files/10185.pdf. (accessed 22 November 2018)
Menocal, Alina. 2015. Inclusive political settlements: evidence, gaps, and challengers of institutional transformation, Governance and Social Development Resource Centre, University of Birmingham, June 2015, Available online: http://publications.dlprog.org/ARM_PoliticalSettlements.pdf. (accessed 21 November 2018).
|
Need to improve the general language of the paper.
Some few specifics are highlighted (red highlights are missing)
1. Line 90 - Within the above context, this paper is interested in the extent that mining corporations … 2. Lines 94-5 - As ‘Full Disclosure’ reports (2015 and 2016) conducted by the Centre for Environmental Rights (CER), a public interest law group, …. 3. Lines 94 – 100 is too long and complex. It can be broken down to make reading simpler.
| The suggested changes have been undertaken and the paper edited. |
The author(s) will help the readers by interpreting the some of the issues discussed in the literature or theoretical section in the analysis and discussions to see where the paper aligns or departs from other studies | Done – the author has now made stronger links the theoretical section with the analysis and discussion. Please refer to these sections. |
Reviewer 2 Report
I was very interested in the paper's title and abstract, but disappointed in the content of the paper. In particular there is little connection between the interview findings, which are presented as long quotations about specific issues at the 'micro' level, and the general analysis of mining in South Africa at the 'macro' level. More conceptual work and structuring the article is needed to tie the empirical findings together with the literature review and conclusions. I don't disagree with the starting point or the conclusions of the paper regarding mining and democracy in South Africa, but the quality of writing and argumentation is not ready for publication.
Author Response
REVIEWER 2 - Comment | Author response |
There is little connection between the interview findings, which are presented as long quotations about specific issues at the 'micro' level, and the general analysis of mining in South Africa at the 'macro' level. More conceptual work and structuring the article is needed to tie the empirical findings together with the literature review and conclusions | More stronger links are made between the empirical findings and the literature review. However, the author initially decided to keep the broader debates and linkages between findings and the macro-level connection in the conclusion and discussion section.
The conclusions presented do made links with the literature and with the general analysis of mining in South Africa. |
Reviewer 3 Report
Title
Mining corporations, democratic meddling and environmental justice in South Africa
Summary of the paper
The paper studies the influence of mining corporations on democracy and government-related decisions at both the state and the local level in post-apartheid South Africa. Through an interview- and narrative-based approach, the paper underlines how relations and collusions between the mining industry and the (local) authorities has influenced mining approval and development, as well as government's attitude in holding mining corporations accountable for the negative effects of the mining activity on ecology and workers' health.
General comment
The paper deals with a highly interesting topic. Indeed, the discussion presented by the author(s) can be generalised to a variety of regional economies worldwide where collusion between the (mining) industry and the (local) government takes place and influences the socio-economic development of individuals and places. I liked the paper a lot, which is, overall, well written, even though some paragraphs are somehow hard to follow. Instead, I am a bit sceptical about the empirical/interview-based section as it is based on a very little number of interviews which are, in addition, realised with anti-mining persons only. This is, in my opinion, the main limitation of the paper. I have only very few suggestions which, I hope, could help the author(s) to improve the work.
Specific comments
(1) Please, check the manuscript carefully to fix writing typos (e.g. "... to ensure that the health ..." at page 1, line 40).
(2) Please, undergo another round of editing as some paragraphs are somehow hard to follow.
(3) I think that the evidence-based discussion suffers from a main limitation, which concerns the one-side position of interviewed persons with respect to the role of mining companies. I acknowledge the impossibility for the author(s) to handle this issue, as government-related individuals did not participate to the study. The author(s) should, at least, discuss explicitly this drawback (even in a footnote or endnote) as the interview-based discussion proposed in the paper does not present views from both sides, and this limits a lot the reliability of the analysis.
(4) Somehow related to the previous point. Would the author(s) specify how individuals were originally identified for the interviews? Were they randomly sorted from a broader set of mining-involved people, or chosen according to some other specific reason? I am sorry to bother the author(s) on such issues, but the paper presents, in my opinion, a strong "political" view on the topic investigated. This is not a problem itself, but the absence of opinions expressed by the pro-mining and institutional/government sides reduces the scientific ground of the paper.
(5) It would be nice to include some empirical evidence in the paper. I am not asking for statistics or econometrics. I would suggest to include (and discuss, of course) some plots and/or tables giving the reader information on the contribution of the mining industry to the economy. For example, figures on employment, GDP, etc. In addition, it would be nice to have a map showing the geographic distribution of mining spots in the region. Would it be possible to exploit statistical sources to discuss links between the mining industry and local communities' poverty/employment dynamics?
Author Response
REVIEWER 3 - Comment | Author response |
The empirical/interview-based section as it is based on a very little number of interviews, which are, in addition, realised with anti-mining persons only. This is, in my opinion, the main limitation of the paper. | Unfortunately, the author tried to secure interviews with mining and government personnel but this proved fruitless. This challenge is reported on and such challengers are noted and accepted in qualitative techniques.
As the author notes under 2.1 - Regrettably attempts to secure interviews from key personnel within the DMR (i.e. Regional Manager Mpumalanga), including the Department of Water Affairs (DWA) (i.e. Chief Director: Legal Services) was ineffective. For St Lucia…Unfortunately, several attempts in 2016 to get interviews from key personnel such as the Secretariat Regional Mining Development and Environmental Committee (RMDEC) KwaZulu-Natal, Co-operative Governance and Traditional Affairs (COGTA), KwaZulu-Natal Wildlife, Naledi Development Consulting (the Fuleni mining consultants) and the Chief Executive Officer from Hluhluwe-iMfolozi game reserve – proved fruitless.
Despite the above challengers, the findings are not limited but quite telling in government and the mining industry not responding to my interview request.
|
(1) Please, check the manuscript carefully to fix writing typos (e.g. "... to ensure that the health ..." at page 1, line 40). | Done |
Please, undergo another round of editing as some paragraphs are somehow hard to follow | Done – editing done to reduce sentences that seem long. Editing has been done for the entire paper. |
I think that the evidence-based discussion suffers from a main limitation, which concerns the one-side position of interviewed persons with respect to the role of mining companies. I acknowledge the impossibility for the author(s) to handle this issue, as government-related individuals did not participate to the study. The author(s) should, at least, discuss explicitly this drawback (even in a footnote or endnote) as the interview-based discussion proposed in the paper does not present views from both sides, and this limits a lot the reliability of the analysis. | As above this limitation has been discussed under section 2.1. |
Would the author(s) specify how individuals were originally identified for the interviews? Were they randomly sorted from a broader set of mining-involved people, or chosen according to some other specific reason? I am sorry to bother the author(s) on such issues, but the paper presents, in my opinion, a strong "political" view on the topic investigated. This is not a problem itself, but the absence of opinions expressed by the pro-mining and institutional/government sides reduces the scientific ground of the paper. | How individuals were identified have been added to section 2.1 under participants. |
It would be nice to include some empirical evidence in the paper. I am not asking for statistics or econometrics. I would suggest to include (and discuss, of course) some plots and/or tables giving the reader information on the contribution of the mining industry to the economy. For example, figures on employment, GDP, etc. In addition, it would be nice to have a map showing the geographic distribution of mining spots in the region. Would it be possible to exploit statistical sources to discuss links between the mining industry and local communities' poverty/employment dynamics? | Although I do not agree with adding the contribution of the mining industry to the economy as that is not the focus of the paper, I have included a table 1 indicating the contribution of the South African mining sector to direct jobs compared to other sectors, including a figure 1 showing the contribution of the mining sector to the South African GDP compared to other sectors.
Unfortunately, it is not possible to show the geographic distribution of mining spots in the region as such maps do not exist and mining geographic locations have not been recorded.
It is also not possible at this stage to exploit statistical sources to discuss links between the mining industry and local communities' poverty/employment dynamics as these do not exist since these have been under researched areas. Inferences are made for mining employment but with the latter concrete figures unavailable.
|